# Prognostic Significance of CSF-1R Expression in Early Invasive Breast Cancer

**DOI:** 10.3390/cancers13225769

**Published:** 2021-11-18

**Authors:** Nazia Riaz, Samantha Burugu, Angela S. Cheng, Samuel C. Y. Leung, Dongxia Gao, Torsten O. Nielsen

**Affiliations:** 1Department of Pathology and Laboratory Medicine, University of British Columbia, Vancouver, BC V6T 1Z7, Canada; nazia.riaz@ubc.ca (N.R.); Sburugu@alum.ubc.ca (S.B.); angelacheng.ubc@gmail.com (A.S.C.); samuel.leung@vch.ca (S.C.Y.L.); dongxia.gao@vch.ca (D.G.); 2Centre for Regenerative Medicine and Stem Cell Research, Aga Khan University, Karachi 74800, Pakistan

**Keywords:** colony-stimulating factor-1 receptor, immunohistochemistry, prognosis, estrogen receptor-positive breast cancer, invasive breast cancer, immune check points, tumor-associated macrophages, CSF-1/ CSF-1R inhibitors

## Abstract

**Simple Summary:**

Experimental evidence suggests that CSF-1/CSF-1R signaling attracts immune cells called macrophages into the tumor microenvironment which promote the capacity of cancer cells to spread. In this study, we investigated CSF-1R protein expression on breast cancer cells and macrophages and their relationship with other immune cells and breast cancer survival. Our results show that cases with high CSF-1R expression on cancer cells, but not macrophages, are associated with inferior survival, particularly in the common hormone receptor-positive breast cancer group, which persists even in the presence of an active anti-tumor host immune response.

**Abstract:**

Colony-stimulating factor-1 receptor (CSF-1R) signaling promotes an immune suppressive microenvironment enriched in M2 macrophages. Given that CSF-1R inhibitors are under investigation in clinical trials, including in breast cancer, CSF-1R expression and association with immune biomarkers could identify patients who derive greater benefit from combination with immunotherapies. TIMER2.0 and bc-GenExMiner v4.7 were used to assess the correlation of *CSF1R* mRNA with immune infiltrates and prognosis. Following a prespecified training–validation approach, an optimized immunohistochemistry assay was applied to assess CSF-1R on carcinoma cells and macrophages on breast cancer tissue microarray series representing 2384 patients, coupled to comprehensive clinicopathological, biomarker, and outcome data. Significant positive correlations were observed between *CSF1R* mRNA and immune infiltrates. High carcinoma CSF-1R correlated with grade 3 tumors >2 cm, hormone receptor negativity, high Ki67, immune checkpoint biomarkers, and macrophages expressing CSF-1R and CD163. High carcinoma CSF-1R was significantly associated with poor survival in univariate and multivariate analyses. Adverse prognostic associations were retained in ER+ cases regardless of the presence of CD8+ T cells. CSF-1R+ macrophages were not prognostic. High carcinoma CSF-1R is associated with aggressive breast cancer biology and poor prognosis, particularly in ER+ cases, and identifies patients in whom biomarker-directed CSF-1R therapies may yield superior therapeutic responses.

## 1. Introduction

Colony stimulating factor-1 receptor (CSF-1R) is a class III receptor tyrosine kinase that is encoded by the *CSF1R* gene, previously called the *c-fms* protooncogene [1]. CSF-1R is normally expressed in a wide range of cells derived from hematopoietic progenitors, including monocytes, tissue-resident macrophages, dendritic cells, and osteoclasts [1]. Engagement of CSF-1R with its ligands, CSF-1 or the more recently identified interleukin-34, causes receptor homodimerization and progressive autophosphorylation of the tyrosine residues on its cytoplasmic tail. The subsequent downstream signaling involves the activation of MEK, PIK3, and PLC-γ2 pathways that regulate the lineage commitment, survival, proliferation, and biological functions of a myriad of cells, including macrophages that constitute the mononuclear phagocyte system [1,2,3].

Preclinical studies have demonstrated a critical role of CSF-1 [4] and CSF-1R [5] in normal breast development. In clinical studies, it has been shown that compared to low expression in the normal breast epithelium, high expression of CSF-1 and CSF-1R is found in the lactating breast epithelium, supporting a potential role of this cytokine–receptor pair in ductal morphogenesis [4]. Aberrant expression of CSF-1 and CSF-1R has been linked with unfavorable outcomes in hematological and solid organ malignancies [6,7,8]. In breast cancer specifically, preclinical studies suggest that this could be attributed to oncogenic paracrine signaling, whereby tumor cell-derived CSF-1 facilitates the recruitment of CSF-1R expressing polarized macrophages into the tumor microenvironment, thereby promoting angiogenesis, tumor cell migration, and invasion [9,10,11,12,13]. Additionally, an autocrine signaling mechanism has also been described in xenograft models of the claudin-low subtype of breast cancer, where TGF-β-driven CSF-1R expression maintains the claudin-low phenotype but promotes invasive potential by suppressing junctional proteins; inhibition of CSF-1R signaling in the same model leads to the upregulation of luminal cytokeratins with a reduction in tumor cell invasiveness but enhanced proliferative capacity [14].

Immune checkpoint inhibitor therapy in combination with chemotherapy has emerged as a promising therapeutic modality for early and metastatic triple-negative breast cancer [15,16,17]. Clinical benefit has been demonstrated with regards to higher rates of pathological complete responses [15,18,19] and a modest increase in objective response rates [16], though improvement in progression-free survival and overall survival has been predominantly achieved in PD-L1-enriched, treatment-naïve metastatic tumors [16,17,20]. Durable responses with immune checkpoint blockade have been reported in diverse tumor types [21]; however, such evidence is limited in breast cancer [16,22]. Primary tumor refractiveness and acquired resistance contribute to inconsistent responses, even in histologically similar tumor types [23]. Factors enabling immune escape are complex and include mechanisms creating an immune-tolerant microenvironment consisting of but not limited to regulatory T cells, myeloid-derived suppressor cells, and tumor-associated macrophages [24].

Dense infiltration with tumor-associated macrophages portends aggressive disease biology and poor prognosis in breast cancer [25]. Tumor-associated macrophages enhance the metastatic potential of carcinoma cells and facilitate remodeling of the extracellular matrix [26]. Equally important is the modulation of the immune microenvironment that contributes to poor treatment responses. A substantial body of experimental evidence indicates that tumor-associated macrophages impair the cytotoxic function of T cells by several mechanisms that include depletion of the metabolic substrate L-arginine via expression of arginase-1, production of reactive oxygen species (which in turn induce expression of PD-L1 on macrophages), and recruiting regulatory T cells [27]. It has also been suggested that TGF-β produced by the macrophages promotes a desmoplastic response in the tumor stroma, which impedes the intratumoral recruitment of cytotoxic T cells [28]. In preclinical models, tumor-associated macrophages contribute to resistance to chemotherapy [29,30], endocrine therapy [31], radiation [32], anti-HER2 [33], and immune-targeted therapies [34,35]. Combination therapies to either deplete or repolarize tumor-associated macrophages have shown encouraging results [36]. In particular, small molecules and monoclonal antibodies targeting CSF-1/CSF-1R signaling have been found to decrease macrophage infiltration and the cancer stem cell pool, enabling the restoration of the antitumor activity of the cytotoxic T cells [37,38]; their clinical efficacy is under investigation in several ongoing clinical trials [36].

Previously, limited studies have reported the prognostic impact of CSF-1R expression in human breast cancers [39,40,41], and the significance of its expression on tumor-associated macrophages in clinical breast cancer specimens is not known. Except for a single study of moderate sample size (*n* = 581) [39], the remaining studies were inadequately powered (fewer than 70 clinical samples each), limiting statistical inferences [40,41]. Given that CSF-1R inhibitors are under active study in clinical trials, including in breast cancer, positive CSF-1R staining in clinical breast cancer specimens and its association with immune cells could help identify which patients might best benefit from combination with immunotherapies. Of particular interest are ER+ breast cancers, where emerging data are suggestive of the presence of a unique microenvironment characterized by tolerant, protumorigenic immune contexture that is rich in tumor-associated macrophages [42]. Hence, the present study was undertaken first to investigate the prognostic significance of *CSF1R* mRNA expression and its correlation with immune cell infiltrates. Subsequently, we applied an in-house optimized CSF-1R immunohistochemistry assay on a particularly large tissue microarray series linked to detailed biomarker and clinical data using a training–validation approach. We sought to test the hypothesis that the expression of CSF-1R protein on carcinoma cells will identify tumors with aggressive features and provide prognostic information in ER+ breast cancers, identifying a subgroup of patients associated with poor clinical outcome. Furthermore, we also determined the prognostic significance of CSF-1R+ tumor-associated macrophages and their association with lymphocytic infiltrates expressing immune checkpoint and CD163+ M2 macrophage biomarkers in the context of ER+ breast cancers.

## 2. Materials and Methods

### 2.1. Bioinformatic Analyses: Correlation of CSF1R mRNA with Stromal Immune Infiltrates, Tumor-Associated Macrophage Gene Signature, and Prognosis

Tumor IMmune Estimation Resource (TIMER) is a web-based portal comprised of three fundamental components, i.e., immune, exploration, and estimation [43,44]. Each of these components has different modules that provide an exhaustive approach for analyzing the tumor-immune interactions across 32 different cancer types from the TCGA database. We accessed the *Gene_Corr* module to investigate the correlation of *CSF1R* mRNA with the 37-gene set described for tumor-associated macrophages [26]. The purity-adjusted partial Spearman’s rho values and associated *p*-values were generated, representing the degree of the correlation. The *Gene* module was used for analyzing the correlation between the *CSF1R* mRNA and immune infiltrate levels through a statistical deconvolution method described previously [45], and the results were displayed as scatter plots with the purity-adjusted partial Spearman’s rho values and the associated *p*-values. 

The Breast Cancer Gene-Expression Miner v4.7, an open-access, web-based tool [46], was utilized to explore the expression of *CSF1R* mRNA in the invasive breast cancer cases represented in the TCGA cohort (*n* = 1033) [47] and validated in the Molecular Taxonomy of Breast Cancer International Consortium (METABRIC) cohort (*n* = 1980) [48]. Targeted analysis was selected, prognostic significance of *CSF1R* mRNA was estimated by applying a univariate Cox proportional hazard model, and results were presented as Kaplan–Meier survival curves for disease-free survival. The median threshold was selected for classifying *CSF1R* mRNA expression levels as high versus low.

### 2.2. Breast Cancer Clinical Cohorts for Optimizing Immunohistochemistry Protocol and Hypothesis Testing 

Assessment of CSF-1R protein was initially performed on a local population series called the UBC cohort to finalize the immunostaining protocol, interpretation, and optimal scoring criteria before applying to the larger BC Cancer cohort. The patients included in the UBC cohort (*n* = 330) were diagnosed with early invasive breast cancer at the University of British Columbia Hospital between 1989 and 2002. The clinicopathological features, surgical procedures, adjuvant radiation/systemic treatments, and clinical outcomes for a median follow up of 13 years have been described previously [49].

The BC cancer cohort represents our independent, main series (*n* = 2384). These patients were diagnosed with stage I–III invasive breast cancer between 1986 and 1992 at the British Columbia Cancer Agency, Vancouver. The adjuvant treatments recommended for these patients were in accord with the provincial guidelines at the time and preceded anti-HER2 and immune checkpoint inhibitor therapies. The median follow-up of the patients is 12.5 years. This cohort has been comprehensively profiled for several key biomarkers, [50] including those related to immune-oncology, i.e., stromal tumor-infiltrating lymphocytes (assessed on hematoxylin- and eosin-stained slides), CD8, FOXP3, TIM3, LAG-3, PD-1 (on intraepithelial tumor-infiltrating lymphocytes (iTILs)), PD-L1 (on carcinoma cells), and CD163 (on macrophages), and assessed by immunohistochemistry [51,52,53,54,55]. The patient population of the BC Cancer cohort has been split into internal training and validation sets as described previously [56]. For the purpose of this study, the training set (*n* = 1183) was used to perform correlative and prognostic analyses for testing our prespecified hypothesis for the biomarker under study. The significant findings were re-tested for confirmation in the validation set (*n* = 1201). Post-hoc analyses were then performed on a prespecified ER+ subgroup using the combined cohort (*n* = 2384) to explore the prognostic association of CSF-1R expression on carcinoma cells in the context of CSF-1R+ macrophages and other immune infiltrates. Patients diagnosed with bilateral breast cancers, ductal carcinoma in situ without an invasive component, and metastatic disease at presentation, or those who received neoadjuvant chemotherapy, were omitted from the analyses. Details about the clinicopathological characteristics, treatments, relapses, and survival have been described previously [50].

The Clinical Research Ethics Board of the University of British Columbia and the Breast Cancer Outcomes Unit of BC Cancer granted authorization for retrieving the archived tumor samples and the corresponding de-identified clinical information for this retrospective biomarker study, which was conducted in accordance with the Reporting Recommendations for Tumor Marker Prognostic Studies (REMARK) guidelines [57]. A flow diagram illustrating the study design is presented as Appendix A.

### 2.3. Tissue Microarrays and Immunohistochemistry 

Archived formalin-fixed, paraffin-embedded tumor tissues from both the cohorts were utilized for construction of tissue microarrays with a 0.6-mm diameter core size, as described previously [49,58]. Briefly, the UBC cohort was represented in 3 tissue microarray blocks consisting of duplicate cores, while the tumors from the BC Cancer validation cohort were assembled into 17 blocks with one core per case. Sections were cut at 4 µm, and immunostaining was performed on a Ventana Discovery XT stainer. Following heat-induced epitope retrieval (Cell Conditioning 1, Ventana Medical Systems, Inc., Oro Valley, AZ, USA) in sodium citrate buffer (pH 6.0) for 10 min, the sections were incubated with anti-human CSF-1R monoclonal antibody (clone SP211; ab183316) at room temperature for 1 h at 1:500 dilution. Sections were then incubated with secondary antibody (UltraMap anti-Rb HRP) for 16 min, and immunoreactivity was visualized using a ChromoMap DAB detection kit (Ventana Medical Systems, Inc.). Human colonic tissue was used as a positive control; for the negative control, the primary antibody was replaced with phosphate-buffered saline. Tissue microarray slides were digitally scanned at 10X objective magnification on the BLISS system (Bacus Laboratories/Olympus America, Lombard, IL, USA) and were visually scored by a breast pathologist (DG) who was blinded to the clinical information. CSF-1R revealed a membranous/cytoplasmic staining pattern. The proportion of the carcinoma cells positively stained was recorded. High expression was defined when ≥10% carcinoma cells stained positive while staining of <10% was considered as low expression (Figure 1A). For tumor-associated macrophages, up to 100 positively stained cells were counted per core, and the threshold was set at 2 (low, <2 and high, ≥2) (Figure 1B). Cores with fewer than 50 invasive cells were excluded from evaluation. Duplicate cores were handled by estimating the mean of the positively stained cells. Images from these slides are available for public access via the website of the Genetic Pathology Evaluation Center (http://www.gpec.ubc.ca/csf1r (accessed on 27 September 2021)). The scoring methodologies for the immune biomarkers have been described previously [51,52,53,54,55].

### 2.4. Statistical Analyses 

SPSS (IBM SPSS Statistic, Version 25.0) was used for statistical analyses. Descriptive statistics were estimated for categorical and continuous variables. A chi-square test or Fisher exact test was used to determine the correlations between CSF-1R expression and clinicopathological characteristics or immune cell infiltrates. The Kaplan–Meier method was applied for the evaluation of cumulative survival probabilities using breast cancer-specific survival (defined as the period between the date of diagnosis and the date of death due to breast cancer) as the predetermined main survival endpoint. All patients who were either alive at the end of the follow-up or died due to non-breast cancer-related causes were censored. Differences in the survival probabilities between the groups were estimated by the log-rank test. For our hypothesis testing using a training–validation approach, the assumptions for the Cox proportional hazard model were violated by the 20-year follow up data. Hence, these analyses were performed by limiting the breast cancer-specific survival time up to 10 years. For multivariate analysis, Cox proportional hazard models were developed for reporting hazard ratios with 95% confidence intervals after correction for traditional clinicopathological characteristics that included age, tumor size, grade, axillary lymph nodal status, and lymphovascular invasion. A *p*-value of <0.05 was regarded as statistically significant. The Bonferroni correction was applied to address the concerns of multiple comparisons during correlative analyses (involving 10 or more variables), making the criterion for statistical significance *p* < 0.005. Kaplan–Meier survival curves for exploratory analyses are presented on the combined cohort of *n* = 2384 patients, for whom 20-year follow-up data was available. 

## 3. Results

### 3.1. Correlation of CSF1R mRNA Expression with Immune Cell Abundance and Prognosis

We first assessed the correlation of *CSF1R* mRNA expression in the tumor microenvironment with the level of immune infiltration using the TIMER2.0 web server [43,44]. Our results showed a significantly positive correlation of *CSF1R* mRNA expression with immune cell abundance comprising B cells, CD8+ T cells, CD4+ T cells, macrophages, neutrophils, and dendritic cells (Figure 2A).

Higher expression of a recently described 37-gene tumor-associated macrophage gene signature, derived from whole-tumor RNA sequencing of 47 breast cancers, has been shown to associate with aggressive subtypes and poor clinical outcomes [26]. We correlated the expression of *CSF1R* with the tumor-associated macrophage gene set and observed a highly significant positive correlation for all 37 genes (Appendix A). These data support the role of *CSF1R* in regulating the biological functions of tumor-associated macrophages as has been shown in previous preclinical studies [12,59] 

No significant prognostic association was observed between *CSF1R* mRNA expression in the bulk tumor tissue and clinical outcomes in the TCGA cohort. This lack of significant association was further validated in the METABRIC cohort of invasive breast cancers (Figure 2B,C). 

The immune cell infiltration was also associated with significant positive correlation with *CSF1* mRNA expression (Appendix A). However, no significant prognostic associations were observed for *CSF1* mRNA in the TCGA and METABRIC cohorts (Appendix A).

### 3.2. Correlation of CSF-1R+ Carcinoma Cells with Clinicopathological Features, Immune Cell Infiltrates, and Prognosis in The BC Cancer Cohort

The locked down immunohistochemistry protocol was applied to the BC Cancer series. The selection of the optimal positivity threshold and the correlative and prognostic associations of CSF-1R on carcinoma cells and macrophages were first assessed in the training cohort (*n* = 1183) of the BC Cancer series (Appendix A) before confirming the results in the validation cohort, presented in this section. 

In all, there were 1201 scorable cases in the validation set of the BC Cancer series, of which 23.1% (277/1201) were positive for CSF-1R expression on carcinoma cells, and high expression was significantly associated with tumor size > 2 cm, grade 3 histology, Ki67 index ≥ 14%, negative hormone receptor status, HER2 positivity, and non-luminal subtypes (Table 1). CSF-1R+ carcinoma cells are significantly associated with stromal tumor-infiltrating lymphocytes; intraepithelial lymphocytic infiltrates expressing CD8, FOXP3, TIM3, LAG3, and PD-1; and PD-L1+ carcinoma cells (*p* < 0.001) (Table 2). In addition, a significantly positive correlation (*p* < 0.001) is also observed between CSF-1R+ carcinoma cells and macrophages expressing CSF-1R and CD163 (Table 2), supporting the plausible existence of a CSF-1/CSF-1R paracrine signaling mechanism as has been suggested in the preclinical studies [9].

Breast cancers with high expressions of CSF-1R on carcinoma cells are associated with significantly poor breast cancer-specific survival compared to tumors with low expression (Figure 3A) (HR 1.63; 95% CI 1.26–2.09; *p* < 0.001). However, the expression of CSF-1R+ macrophages did not reveal any significant prognostic associations in either the training (Appendix A) or validation sets of the BC Cancer series (HR 1.0, 95% CI 0.80–1.27; *p* = 0.97) (Figure 3B).

We next performed a prespecified subgroup analysis to investigate the prognostic impact of CSF-1R+ carcinoma cells in cases stratified by ER status. We found that CSF-1R+ carcinoma cells are associated with significantly adverse outcome in patients with ER-positive status (HR 1.82, 95% CI 1.30–2.54; *p* < 0.001) but not in those with ER-negative status (HR 1.08, 95% CI 0.73–1.60; *p* = 0.71) (Figure 3C,D). The subsequent multivariate analysis confirms that high CSF-1R expression on carcinoma cells is independently associated with unfavorable outcome (HR 1.58; 95% CI 1.21–2.05; *p* = 0.001) (Table 3). 

### 3.3. Exploratory Analyses: Prognostic Association of CSF-1R-Expressing Carcinoma Cells with Immune Infiltrates in ER-Positive Breast Cancers 

In contrast to triple-negative breast cancers, immune-enriched ER-positive breast cancers are characterized by a distinctive immune profile which is dominated by TGF-β signaling, dysfunctional CD8+ T cells, and enrichment of M2 tumor-associated macrophages [42]. Clinical studies have shown that high levels of FOXP3- and CD8-positive lymphocytic subsets are linked with inferior clinical outcomes in ER-positive breast cancers [52,60]. Here, we performed an exploratory analysis to examine the prognostic impact of CSF-1R+ carcinoma cells in the context of iTIL subsets expressing CD8 or FOXP3 in ER-positive breast cancers in the combined cohort of the BC Cancer series with complete 20-year follow-up information (*n* = 1666). Our results show that high CSF-1R expression on carcinoma cells is associated with significantly inferior survival in cases harboring CD8+ iTILs (HR 1.65, 95% CI 1.21–2.26; *p* = 0.002) (Figure 4A). When analyzed for regulatory T cells, the cases with concurrent low CSF-1R+ carcinoma cells and low FOXP3 iTILs exhibit the best outcome (HR 0.71, 95% CI 0.60–0.86; *p* = 0.001) (Figure 4B). Together, these results suggest an association of CSF-1R+ carcinoma cells in supporting an immune-suppressed microenvironment.

We and others have previously shown that CD163+ M2 macrophages are associated with poor prognosis in breast cancer [25,54]. We performed an exploratory analysis to examine the relationship of carcinoma cells expressing CSF-1R in the context of tumor-associated macrophages stratified by ER expression. We observed that in ER-positive cases, concomitant high expression of CSF-1R on carcinoma cells and macrophages is associated with significantly poorer clinical outcome (HR 1.57, 95% CI 1.13–2.17; *p* = 0.006), compared to ER-negative cases (HR 1.09, 95% CI 0.75–1.57; *p* = 0.66) (Figure 5A,B). Similar results are observed when analysis is performed in CD163+ M2 macrophage-enriched ER-positive breast cancers (Figure 5C,D). Together, these results show a negative prognostic association linked with the concurrent expression of CSF-1R+ carcinoma cells and CSF-1R+ macrophages in ER-positive breast cancers.

## 4. Discussion

We report the prognostic significance of CSF-1R protein expression on carcinoma cells and tumor-associated macrophages in a large, well-characterized cohort of early breast cancer patients. Key findings of our study include (a) poor clinical outcome associated with high expression of CSF-1R+ carcinoma cells that was maintained in ER-positive breast cancers despite the presence of infiltrating lymphocytes expressing CD8 and (b) a significant positive correlation between CSF-1R+ carcinoma cells and tumor-associated macrophages in a manner that suggests that its expression on carcinoma cells could be a factor driving poor outcome in the ER-positive subgroup, whereas a lack of its expression on malignant epithelial cells corresponds with significantly improved outcome regardless of the presence of CSF-1R+ tumor-associated macrophages.

Our analysis of TCGA and METABRIC data sets in contrast did not reveal a significant prognostic association for *CSF1R* or *CSF1* at the mRNA level. These observations are not necessarily surprising. Since bulk tumor-derived transcriptomic data quantify total gene expression from heterogenous cell types, *CSF1R* and *CSF1* transcripts could be mostly derived from the stromal tumor-associated macrophages (which express this highly in comparison to relatively low expression in carcinoma cells); *CSF1*, in addition, is also expressed by several stromal cells such as fibroblasts, endothelial cells, and smooth muscle cells [61]. Our findings in this regard are important, as using immunohistochemistry, we were able to distinguish between carcinoma cell and macrophage cell CSF-1R expression. We believe that our data, showing a lack of prognostic association of CSF-1R on macrophages, are in agreement with the results of transcriptomic analysis on the TCGA and METABRIC data sets (Figure 2B,C).

Previously, few studies [39,40,41,62] have investigated CSF-1R expression in clinical samples showing prognostic relevance with regards to ipsilateral breast cancer recurrence [62] and nodal involvement [39]. However, these studies, being underpowered, either failed to show statistically meaningful results [40] or could not confirm the independent prognostic significance of CSF-1R beyond standard clinical factors [39].

The oncogenic role of CSF-1R has been described mostly in relation to its ligand, CSF-1. However, a ligand-independent protumorigenic role for CSF-1R has also been shown in preclinical models, where activating mutations in the *c-fms* protooncogene have shown transforming potential in NIH 3T3 cells independent of the CSF-1 ligand [63]. Likewise, using an MCF10A cell line model, it has been shown that constitutively active CSF-1R can alter acinar morphology by inducing destabilization of cell-to-cell adhesions via a mechanism that involves src kinases [64]. Together, these data suggest that constitutively active CSF-1R may potentially drive oncogenesis. However, these observations warrant further investigation in clinical breast cancer specimens.

Our immunohistochemistry results are particularly relevant in the context of ER-positive breast cancers, which comprise approximately 70% of all newly diagnosed invasive breast cancers [65]. Whether CSF-1R signaling predominantly regulates the luminal phenotype needs further investigation. However, preclinical studies do suggest that CSF-1R signaling may have a role in regulating luminal differentiation, as mice bearing homozygous mutations in *CSF1R* have defects in mammary ductal branching [5], and that hyperstimulation of CSF-1R disrupts the formation of ductal acini, inducing hyperproliferation in a non-malignant luminal cell line model [64].

Overall, ER-positive breast cancers represent indolent tumors that are generally considered less immunogenic on account of low tumor mutational burden [66], relatively lower expression of immune checkpoints [67], and generally low counts of tumor-infiltrating lymphocytes [68,69]. Emerging data challenge this paradigm and support the existence of distinct immune subsets in ER-positive breast cancers [70]. Recently, it has been shown that, in contrast to immune-rich triple-negative breast cancers, immune-rich ER-positive breast cancers are characterized by TGF-β signaling, which dampens host anti-tumor immune responses by promoting the exhaustion of CD8+ T cells, infiltration with FOXP3+ T regulatory subsets, and M2 macrophage polarization [42]. Together, these data provide some insight into the heterogenous responses of advanced stage ER-positive breast cancers to immune checkpoint inhibition in clinical trials [71,72,73,74]. Whilst prior treatments may possibly weigh heavily in producing sub-optimal responses, the use of immune checkpoint inhibitors in the neoadjuvant setting have yielded encouraging results in hormone receptor-positive/HER2-negative breast cancers [19].

Given the high frequency of M2 macrophages in ER-positive breast cancers [42] and their plausible role in inducing resistance to standard adjuvant and immune checkpoint inhibitor therapies [29,31,32,33,34], combination therapies, including those targeting tumor-associated macrophages in breast cancers, are an attractive concept. Preclinical studies have shown that targeting CSF-1/CSF-1R signaling depletes tumor-associated macrophages by inducing apoptotic death [75], inhibits their recruitment from circulating monocytes [76], and induces M2–M1 polarization [34]. The net result is a reprogrammed immune contexture with a switch from immune-suppressed to an immune-activated, tumoricidal microenvironment [34,59]. While results from trials testing combined targeting of PD-1/LI and CSF-1R are still pending [37], promising efficacy signals of CSF-1R inhibitors have been demonstrated in combination with chemotherapy in solid organ malignancies, including breast cancer [77]. Of note, despite their capacity to eliminate tumor-associated macrophages, limited anti-cancer activity has been observed with use of CSF-1R inhibitors alone or in combination with chemotherapeutic agents [78,79]. Indeed, these observations are in part supported by preclinical studies showing a significant impact of the timing of initiation of anti-CSF-1R therapies, showing diminished anti-tumor activity when CSF-1R inhibitors are administered in mice bearing established tumors versus when therapy is started concurrent with tumor implantation [80]. It has been shown that the carcinoma cells actively contribute to compensatory mechanisms that promote the survival of macrophages following CSF-1R inhibitor therapy [81].

Even though several biomarkers for tumor-associated macrophages have shown prognostic implications [25], at present, no companion diagnostic tests are available to predict benefit from macrophage-targeted interventional therapies. Our biomarker study demonstrates that CSF-1R expression on tumor-associated macrophages does not provide prognostic value; however, it is likely that in vivo CSF-1R expression on both the carcinoma cells and host macrophages contributes to the pharmacologic effects of inhibitors. Whether combined assessment of CSF-1R expression on both the carcinoma cells and immune cells demonstrates better predictive capacity needs further investigation in correlative studies linked to trials of CSF-1R-targeted therapies. 

Strengths of our study include the utilization of a hypothesis-driven training–validation approach for evaluation of CSF-1R expression and the large size of the BC Cancer cohort, for there exist previously generated extensive data relevant to immune biomarkers. Our study has some limitations which must be acknowledged. Firstly, our study was designed to evaluate the expression of CSF-1R on carcinoma cells and tumor-associated macrophages using a single biomarker immunohistochemistry. However, the CSF-1R signaling axis also influences other immune and non-immune subsets, including dendritic cells, neutrophils, and cancer-associated fibroblasts, which may either augment or neutralize the effects of CSF-1R inhibition [80,82]. Comprehensive evaluation of the tumor microenvironment with multiplexed approaches might improve prognostic and predictive capacity, particularly in the context of immune checkpoint inhibitor therapies [83]. Secondly, we did not evaluate the immunohistochemical expression of CSF-1, a major ligand of CSF-1R that influences the recruitment and polarization of M2 tumor-associated macrophages. Thirdly, we assessed CSF-1R expression on morphologically assessed tumor-associated macrophages and did not perform multiplex immunohistochemistry to confirm co-expression with other macrophage biomarkers. Fourthly, our study was based on immunohistochemistry performed on formalin-fixed tissues assembled on tissue microarray cores which were sampled from a tumor-rich region and hence may not sufficiently capture the heterogeneity of the immune microenvironment [84]. Lastly, the patients included in our cohorts received treatments that were approved by provincial guidelines that date back to more than 30 years and do not reflect the more contemporary adjuvant treatments that are presently recommended.

## 5. Conclusions

Our results demonstrate that CSF1-R expression on carcinoma cells detected by immunohistochemistry provides prognostic information in breast cancers, particularly in a subset of ER-positive cases, and may represent an actionable target for patient selection in trials evaluating CSF-1R blockade. Findings from our exploratory analyses showing a negative association of CSF-1R+ carcinoma cells even in immune-rich ER-positive breast cancers require validation in independent clinical cohorts.

## Figures and Tables

**Figure 1 cancers-13-05769-f001:**
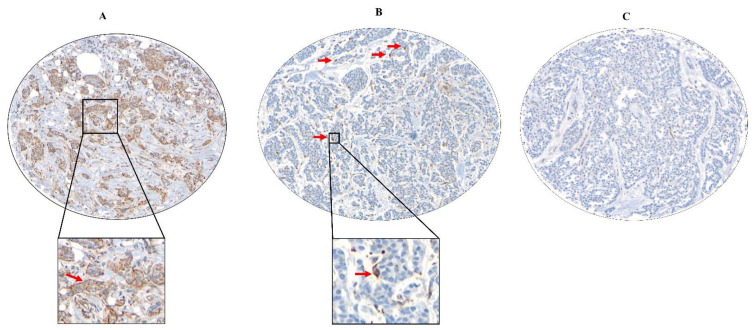
Representative photomicrographs for immunohistochemical staining of CSF-1R expression in breast carcinoma tissue microarray cores. (**A**) Cytoplasmic/membranous expression of high CSF-1R showing ≥10% positivity on carcinoma cells; (**B**) cytoplasmic/membranous expression of CSF-1R on tumor-associated macrophages (red arrows) showing ≥2 positive macrophages; (**C**) negative expression on tumor cells and macrophages (images acquired at 10×). The inserts show CSF-1R+ carcinoma cells (**A**) and CSF-1R+ tumor-associated macrophages (**B**) at high magnification.

**Figure 2 cancers-13-05769-f002:**
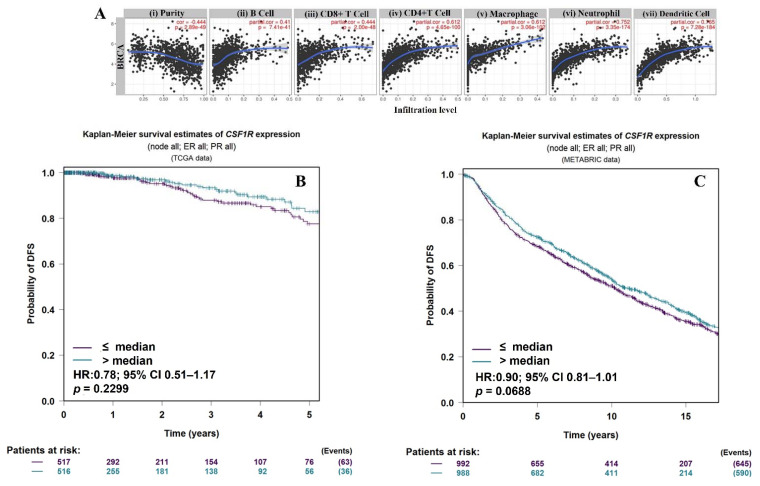
Correlation of *CSF1R* mRNA expression with immune cell infiltrates and prognosis in breast cancer. TIMER2.0 web server was used to evaluate the correlation between *CSF1R* mRNA expression and immune cell infiltrates. The left-most panel (**A**(i)) shows *CSF1R* mRNA expression against tumor purity (proportion of cancer cells in the sample), showing a significantly negative correlation. Scatter plots (**A**(ii–vii)) show the purity-adjusted partial Spearman’s rho value and statistical significance for positive correlation of *CSF1R* mRNA with immune cell types (Spearman’s rho: 0, no correlation; 0.1–0.3, weak; 0.4–0.6, moderate; 0.7–0.9, strong; 1, perfect; *p*-value < 0.05). Five-year Kaplan–Meier curves show an insignificant association of *CSF1R* mRNA with disease-free survival (DFS) in the TCGA (**B**) and METABRIC (**C**) invasive breast cancer cohorts. The *p*-value and hazard ratios (HR) with corresponding 95% confidence intervals were estimated by the log-rank test. The survival curves for TCGA and METABRIC cohorts were generated using Breast Cancer Gene-Expression Miner v4.7.

**Figure 3 cancers-13-05769-f003:**
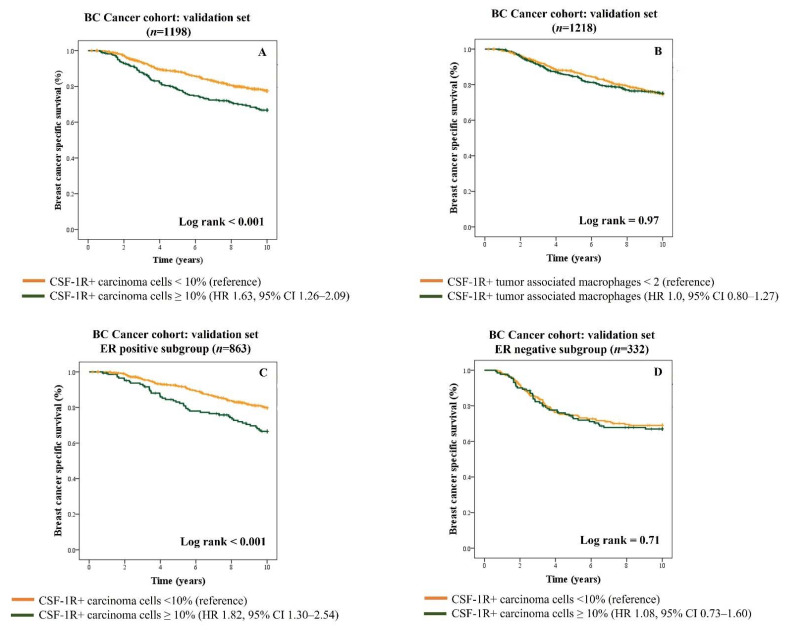
Kaplan–Meier survival curves showing breast cancer-specific survival in the validation set of BC Cancer series stratified by CSF-1R+ carcinoma cells (**A**) and CSF-1R+ tumor-associated macrophages (**B**). Prespecified subgroup analysis for CSF-1R-expressing carcinoma cells in ER-positive cases (**C**) and ER negative cases (**D**).

**Figure 4 cancers-13-05769-f004:**
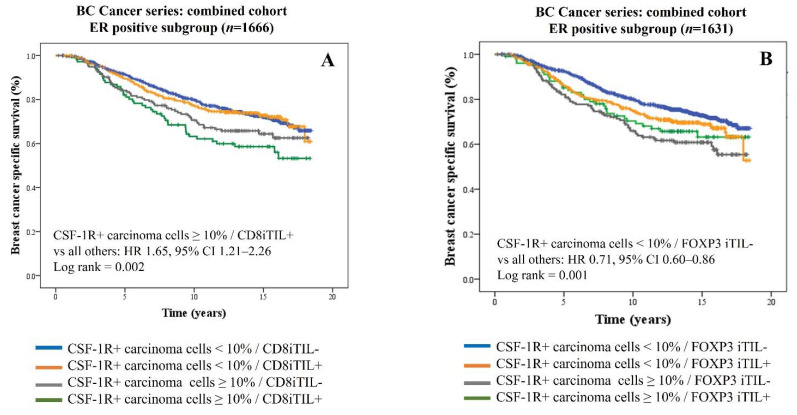
Kaplan–Meier survival curves for breast cancer-specific survival in ER-positive cases in the full BC Cancer cohort stratified by CSF-1R expression on carcinoma cells and intraepithelial tumor-infiltrating lymphocytes (iTILs) expressing CD8 (**A**) or FOXP3 (**B**).

**Figure 5 cancers-13-05769-f005:**
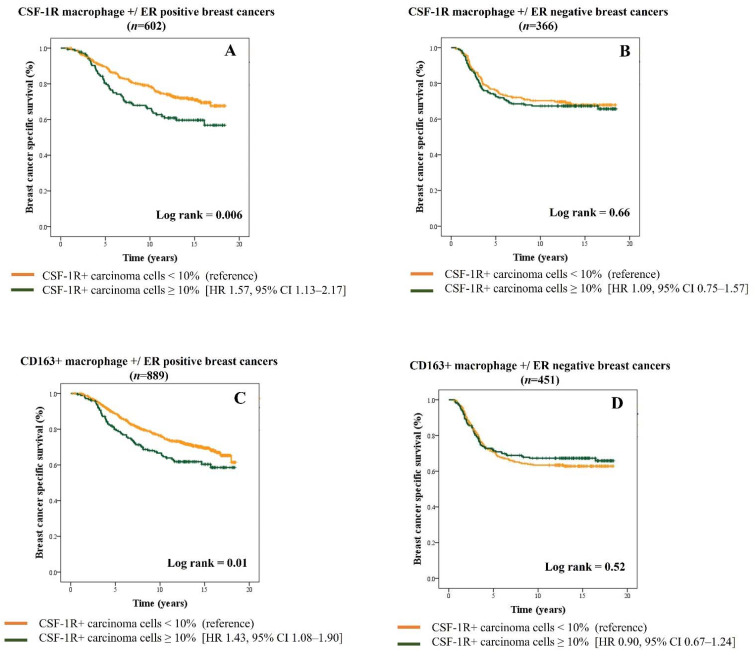
Kaplan–Meier survival curves showing breast cancer-specific survival in combined BC Cancer series in the presence of CSF-1R+ macrophages stratified by CSF-1R+ carcinoma cells in ER-positive (**A**) and ER-negative cases (**B**). Kaplan–Meier curves in the presence of CD163+ macrophages stratified by CSF-1R+ carcinoma cells in ER-positive (**C**) and ER-negative cases (**D**).

**Table 1 cancers-13-05769-t001:** Validation set of BC Cancer cohort: correlation of CSF-1R+ carcinoma cells with clinicopathological features.

Clinicopathological Variables	CSF-1R Expression on Carcinoma Cells		*p*-Value
	Low (<10%) *n* = 924 (76.9%)	High (≥10%) *n* = 277 (23.1%)	
**Age at diagnosis**			0.57
<50	307 (33.2)	87 (31.4)	
≥50	617 (66.8)	190 (68.6)	
**Tumor size (cm)**			0.001 *
≤2	484 (52.7)	113 (41.1)	
>2	434 (47.3)	162 (58.9)	
**Tumor grade**			<0.001 *
1 and 2	404 (45.4)	86 (31.9)	
3	486 (54.6)	184 (68.1)	
**Axillary lymph node status**			0.11
Negative	519 (56.2)	140 (50.7)	
Positive	404 (43.8)	136 (49.3)	
**Lymphovascular invasion**			0.37
Negative	459 (51.2)	144 (54.3)	
Positive	437 (48.8)	121 (45.7)	
**ER expression**			<0.001 *
Negative	201 (21.8)	131 (47.3)	
Positive	720 (78.2)	146 (52.7)	
**Progesterone receptor expression**			<0.001 *
Negative	392 (44.7)	169 (63.5)	
Positive	484 (55.3)	97 (36.5)	
**HER2 overexpression/amplification**			<0.001 *
Negative	816 (89.9)	203 (74.9)	
Positive	92 (10.1)	68 (25.1)	
**Ki-67 proliferation index**			<0.001 *
<14%	502 (58.3)	92 (35.8)	
≥14%	359 (41.7)	165 (64.2)	
**Breast cancer subtypes**			<0.001 *
Luminal NOS	37 (4.0)	5 (1.8)	
Luminal A	438 (47.4)	58 (20.9)	
Luminal B/HER2-/Ki67+	217 (23.5)	64 (23.1)	
Luminal/HER2+	47 (5.1)	20 (7.2)	
HER2+	44 (4.8)	45 (16.2)	
Basal	66 (7.1))	53 (19.1)	
Additional basal by TNP	50 (5.4)	18 (6.5)	
Unassignable	25 (2.7)	14 (5.1)	

* Denotes differences between low and high CSF-1R groups that are significant at the Bonferroni-corrected *p*-value of <0.005 (0.05/10); ER, estrogen receptor; HER2, human epidermal growth factor receptor; TNP, triple-negative phenotype.

**Table 2 cancers-13-05769-t002:** Validation set of BC Cancer cohort: correlation of CSF-1R+ carcinoma cells with immune biomarkers.

Variables	CSF-1R Expression on Carcinoma Cells		*p*-Value
	Low (<10%)	High (≥10%)	
**H&E sTIL count (%)**			<0.001
<10	723 (84.8)	194 (73.8)	
≥10	130 (15.2)	69 (26.2)	
**CD8 iTIL count**			0.001
<1	602 (67.7)	149 (56.4)	
≥1	287 (32.3)	115 (43.6)	
**PD-1 iTIL count**			<0.001
<1	809 (91.7)	219 (81.7)	
≥1	73 (8.3)	49 (18.3)	
**PD-L1+ carcinoma cells (%)**			<0.001
0	814 (94.1)	215 (81.4)	
≥1	51 (5.9)	49 (18.6)	
**FOXP3 iTIL count**			<0.001
<2	624 (70.1)	141 (52.8)	
≥2	266 (29.9)	126 (47.2)	
**TIM3 iTIL count**			<0.001
<1	804 (90.5)	212 (78.5)	
≥1	84 (9.5)	58 (21.5)	
**LAG3 iTIL count**			<0.001
<1	793 (89.5)	208 (77)	
≥1	93 (10.5)	62 (23)	
**CD163+ M2 macrophages**			<0.001
≤5	336 (40.5)	61 (24.2)	
>5	493 (59.5)	191 (75.8)	
**CSF-1R+ M2 macrophages**			<0.001
<2	532 (61.2)	108 (39.6)	
≥2	337 (38.8)	165 (60.4)	

H&E, hematoxylin- and eosin-stained; iTILs, intraepithelial tumor infiltrating lymphocytes; sTILs, stromal tumor-infiltrating lymphocytes; PD/L1, programmed cell death protein/ligand 1; FOXP3, forkhead box P3; TIM3, T cell immunoglobulin domain and mucin domain 3; LAG3, lymphocyte activation gene 3 protein; CSF-1R, colony-stimulating factor-1 receptor.

**Table 3 cancers-13-05769-t003:** Multivariate analysis for CSF-1R+ carcinoma cells in the validation set of BC Cancer cohort.

Co-Variates in the Model	Breast Cancer Specific Survival	
	Adjusted HR (95% CI)	*p*-Value
**Age at diagnosis (years)**		0.70
<50	1	
≥50	1.05 (0.81–1.36)	
**Tumor size (cm)**		<0.001
≤2	1	
>2	1.65 (1.26–2.15)	
**Tumor grade**		<0.001
1 and 2	1	
3	2.03 (1.52–2.71)	
**Axillary LN status**		<0.001
Negative	1	
Positive	1.84 (1.38–2.44)	
**LVI**		0.02
Negative	1	
Positive	1.40 (1.05–1.88)	
**CSF-1R+ carcinoma cells**		0.001
Low (<10%)	1	
High (≥10%)	1.58 (1.22–2.05)	

LN, lymph node; LVI, lymphovascular invasion; CSF-1R, colony-stimulating factor-1 receptor.

## Data Availability

As per the institutional policy for protecting patient privacy, the clinical information included in this study is not publicly available. Requests or inquiries for accessing the clinical data files should be referred to the corresponding author. The data presented in this study can be made available to qualified researchers only if a satisfactory requisition is approved by the Breast Cancer Outcomes Unit of BC Cancer, subject to completion of a Data Transfer Agreement and confirmation of ethical approval. Images from the tissue microarray slides are available for public access via the website of Genetic Pathology Evaluation Center (http://www.gpec.ubc.ca/csf1r (accessed on 27 September 2021)).

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
