# Peer review of "Prognostic Significance of CSF-1R Expression in Early Invasive Breast Cancer"

_cancers, 2021, doi:10.3390/cancers13225769_

Round 1
Reviewer 1 Report
Diaz et al made a sound analyisis regarding the prognostic marker CSF-1R.
I rarely read such straight forward and well made paper and such rigorous statistical analysis where the hypothesis is given and tested throughly.
I have only minor spelling mistakes which needs to be corrected and please avoid data not shown sentences.
Line 217 - 219 please rephrase the sentence. Patients are not "expire".
Line 251 please provide the lymphocyte status analysis as a supplementarry.
Table 1 lymphovascular invasion you have two dots in the table.
Reviewer 2 Report
In the work of Riaz and colleagues, the predictive value of CSFR-1 expression was assessed in invasive breast cancer, combing bioinformatic approaches and IHC validation. The potential correlation with immune cell infiltrates could be a promising approach to identify patients that can gain higher benefits from immunotherapy protocols.
Mayor Issues
- The key weak point of the work is that the CSF1/CSFR1 axis has not been investigated. The central hypothesis is that CSFR1+ macrophages are recruited by the tumor, thus promoting angiogenesis, immune escape, metastasis, etc. However, the authors failed to provide evidence about it. CSFR1 expression on macrophages was not associated with differences in patient prognosis or other immune cell infiltration (in both bioinformatic analyses and IHC validation). CSF1 expression by tumor cells or stromal cells has not been investigated to justify macrophage recruitment.
- The expression of CSFR1 on tumor cells is predictive of breast cancer prognosis. This is a promising result (but not so novel) that deserves to be further explored. The authors claim that “These data support the role of CSFR1 in regulating the biological functions of tumor-associated macrophages (line 244)” is not actually supported by their results. How is it possible to explain the connection between the CSFR1 receptor expressed on tumor cells/macrophages and macrophage recruitment? The interaction between the two populations is likely to be a paracrine signaling. It would be better to associate CSF1 (the ligand) with macrophage recruitment, as suggested above. Probably, here CSFR1 is involved in a different pro-tumorigenic mechanism which is completely missed by the authors.
- When FOXP3+ TILs were analyzed, the authors failed to consider that FOXP3 is expressed also on T-regulatory cells (CD4+) which actually are the principal FOXP3+ TIL population in tumors. I guess the authors did not discriminate between the two populations (FOXP3 CD8 vs FOXP3 CD4 T cells).
Reviewer 3 Report
The manuscript entitled “Prognostic Significance of CSF-1R Expression in Early Invasive 2 Breast Cancer” demonstrated the adverse effect of CSF-1R expression by cancer cells on breast cancer patient prognosis. The authors begin the analysis exploiting available data provided by TCGA and METABRIC data sets. The main conclusions are drawn based on the analysis of CSF-1R protein expression by performing immunohistochemistry on a large cohort of well-characterized breast cancer samples. Overall, the study is well-designed and the manuscript is well-written. The results are clearly presented and the data obtained have a potential translational impact, especially when taking into consideration the ongoing clinical trials targeting CSF-1/CSF-1R axis. Few comments are provided below that could further improve manuscript quality.
In Figure 1 the authors refer that CSF-1R is expressed by tumor associated macrophages. This point needs to be revised throughout the result section, since double/multiple immunostaining is required to demonstrate the expression of CSF-1R by macrophages. Along this line, the authors could assess CSF-1R positive immunostaining separately in the cancer cells and the surrounding stromal cells.
In Figure 1, inserts with higher magnification denoting CSF-1R expression in on stromal cells could improve Figure quality.
Round 2
Reviewer 2 Report
I would like to thank the authors for their responses.I still have a few points to address before accepting the paper.
1) In their response, the authors clarified that "a ligand independent protumorigenic role for CSF-1R has also been shown in preclinical models where activating mutations in the c-fms protooncogene have shown transforming potential in NIH 3T3 cells independent of the CSF-1 ligand." This concept must be added to the discussion, because it is a key point that CSF-1R has both CSF1 ligand-dependent and -independent functions, thus explaining some of their results.
2) Please correct the ambiguous sentence (line 319). "iTILs subsets expressing CD8 and FOXP3" should be changed into "iTILs subsets expressing CD8 or FOXP3", since FOXP3+ T regulatory cells were analyzed by single biomarker immunohistochemistry on the BC, as specified by the authors.
3) I agree with the authors that IHC for CSF1 is not a reliable method to quantify a soluble factor. However, I warmly suggest including at least some of the Bioinformatic analyses to confirm (or not) the correlation of CSF1 mRNA expression with the level of immune infiltration through TIMER2.0 and/or the prognostic value of CSF1 in the TCGA cohort.
